# Cerebral Iron Deposition in Neurodegeneration

**DOI:** 10.3390/biom12050714

**Published:** 2022-05-17

**Authors:** Petr Dusek, Tim Hofer, Jan Alexander, Per M. Roos, Jan O. Aaseth

**Affiliations:** 1Department of Neurology and Center of Clinical Neuroscience, 1st Faculty of Medicine and General University Hospital in Prague, Charles University in Prague, 120 00 Prague, Czech Republic; 2Department of Radiology, 1st Faculty of Medicine and General University Hospital in Prague, Charles University in Prague, 120 00 Prague, Czech Republic; 3Department of Environmental Health, Division of Climate and Health, Norwegian Institute of Public Health, P.O. Box 222, Skøyen, 0213 Oslo, Norway; tim.hofer@fhi.no (T.H.); jan.alexander@fhi.no (J.A.); 4Institute of Environmental Medicine, Karolinska Institutet, 171 77 Stockholm, Sweden; per.roos@ki.se; 5Department of Clinical Physiology, St. Goran Hospital, 112 81 Stockholm, Sweden; 6Research Department, Innlandet Hospital Trust, P.O. Box 104, 2381 Brumunddal, Norway; jaol-aas@online.no; 7Faculty of Health and Social Sciences, Inland Norway University of Applied Sciences, P.O. Box 400, 2418 Elverum, Norway

**Keywords:** neurodegeneration, iron accumulation, chelation, ferroptosis, MRI, NBIA, siderosis

## Abstract

Disruption of cerebral iron regulation appears to have a role in aging and in the pathogenesis of various neurodegenerative disorders. Possible unfavorable impacts of iron accumulation include reactive oxygen species generation, induction of ferroptosis, and acceleration of inflammatory changes. Whole-brain iron-sensitive magnetic resonance imaging (MRI) techniques allow the examination of macroscopic patterns of brain iron deposits in vivo, while modern analytical methods ex vivo enable the determination of metal-specific content inside individual cell-types, sometimes also within specific cellular compartments. The present review summarizes the whole brain, cellular, and subcellular patterns of iron accumulation in neurodegenerative diseases of genetic and sporadic origin. We also provide an update on mechanisms, biomarkers, and effects of brain iron accumulation in these disorders, focusing on recent publications. In Parkinson’s disease, Friedreich’s disease, and several disorders within the neurodegeneration with brain iron accumulation group, there is a focal siderosis, typically in regions with the most pronounced neuropathological changes. The second group of disorders including multiple sclerosis, Alzheimer’s disease, and amyotrophic lateral sclerosis shows iron accumulation in the globus pallidus, caudate, and putamen, and in specific cortical regions. Yet, other disorders such as aceruloplasminemia, neuroferritinopathy, or Wilson disease manifest with diffuse iron accumulation in the deep gray matter in a pattern comparable to or even more extensive than that observed during normal aging. On the microscopic level, brain iron deposits are present mostly in dystrophic microglia variably accompanied by iron-laden macrophages and in astrocytes, implicating a role of inflammatory changes and blood–brain barrier disturbance in iron accumulation. Options and potential benefits of iron reducing strategies in neurodegeneration are discussed. Future research investigating whether genetic predispositions play a role in brain Fe accumulation is necessary. If confirmed, the prevention of further brain Fe uptake in individuals at risk may be key for preventing neurodegenerative disorders.

## 1. Introduction

Disruption of cerebral iron (Fe) regulation appears to have a role in the pathogenesis of various neurodegenerative disorders, where Fe accumulates in different brain structures. Possible effects of this Fe deposition include degeneration of neurons and glial cells and neuroinflammation with infiltration of immune cells into sites of minor lesions [1,2]. Deposition of Fe and concomitant neuroinflammation can be caused by release of heme following vascular hemorrhages, by dysregulations of Fe transport proteins or other regulatory molecules [3], or by increased environmental Fe exposure [4]. The field of therapeutic use of Fe chelators in neurodegenerative diseases is still evolving. Advances in magnetic resonance imaging (MRI) techniques such as quantitative susceptibility mapping (QSM) or R2* relaxometry have enabled quantification of Fe content in specific brain regions. Using MRI, it is now possible to longitudinally monitor changes in cerebral Fe content during disease progression as well as during chelation therapy in vivo. Determination of metal-specific content inside individual cell-types, sometimes also within specific cellular compartments is now possible with modern analytical methods ex vivo. The primary aim of the present review is to summarize the whole brain, cellular, and subcellular patterns of iron accumulation in neurodegenerative diseases of genetic and sporadic origin. We also give an update on mechanisms, biomarkers, and effects of brain iron accumulation including novel emergent therapies.

## 2. Brain Iron, Neuroinflammation, and Cognitive Decline in Aging

Within the brain, Fe is the most abundant transition metal, vital for numerous cellular processes including neurotransmitter synthesis, myelination, and mitochondrial function. There is no Fe in the brain at birth [5]; it increases rapidly between youth and mid-age to remain relatively constant thereafter [6]. Fe accumulation with age predominantly takes place in basal ganglia and other brain regions concerned with motor functions. During adult lifespan, the steepest accumulation is observed in the red nucleus, substantia nigra (SN), and putamen [7,8]. Whole-brain MRI studies showed that, in addition to the deep gray matter, precentral, prefrontal, and occipital cortices involved in motor, cognitive, and visual function also accumulate iron with aging [7,9] (Table 1). Glial cells (astrocytes, oligodendrocytes, microglia) contain higher Fe concentrations than neurons whereby astrocytes are the primary Fe homeostatic cells in the central nervous system. A recent study in young rats found that the Fe concentration of neocortical oligodendrocytes is fivefold higher, of microglia threefold higher, and of astrocytes twofold higher, than in neurons [10]. Another study comparing 2-, 6-, 19-, and 27-month old mice found that not only Fe, but also copper (Cu) and zinc (Zn) concentrations, together with microglia and astrocyte numbers, increased with age in the basal ganglia [11]. In aged human brains, Fe staining intensity increases mostly in astrocytes and microglia [12].

The minute cellular free Fe pool is tightly regulated, as labile Fe catalyzes formation of toxic reactive oxygen species (ROS), e.g., the superoxide anion (O_2_^•−^; weak oxidant) which dismutates into hydrogen peroxide (H_2_O_2_; strong oxidant). H_2_O_2_ mediated oxidation of biomolecules (Fenton chemistry) can occur through a one-electron transfer involving production of the toxic hydroxyl radical (HO^•^; highly reactive, reacts by additions to biomolecules in a random fashion) or a two-electron transfer with reduction and detoxification of H_2_O_2_ [13,14]. Mitochondria in aged tissue can be especially prone to oxidative stressors (e.g., chemicals interacting with the respiratory chain, leading to ROS formation, e.g., O_2_^•−^), but may not necessarily produce more ROS in aging [15] as damaged mitochondria are removed by autophagy [16]. Iron and ROS generation are involved in ferroptosis, a recently described form of cell death which is defined as iron-dependent regulated necrosis that is caused by massive lipid peroxidation-mediated membrane damage [17,18]. Ferroptosis can be triggered by several coherent ways: (i) inactivation of cysteine/glutathione (GSH) antiporter system, (ii) decreased activity of glutathione peroxidase 4 (GPX4), (iii) excessive lipid peroxidation and its end-products such as 4-hydroxynonenal (4-HNE) and malondialdehyde (MDA), and (iv) cellular iron accumulation [19]. The key regulators of ferroptosis are GPX4, a selenoenzyme that catalyzes the reduction of lipid peroxides in a GSH-dependent reaction and nuclear factor erythroid 2-related factor 2 (NRF2), a transcription factor that controls many key components of the ferroptosis pathway [20]. Antioxidants (e.g., vitamin E, selenoenzymes (i.a. GPX4), coenzyme Q10—the main lipid soluble antioxidant) and other repair systems are critical to limit ferroptosis and maintain neuronal function with age [21,22].

The main Fe buffering and storage protein ferritin is found in glial cells but generally not in neurons [23]. Catecholaminergic neurons can to some extent in a slow process bind excess Fe through neuromelanin (NM) formation, a dark-brown Fe-binding pigment formed non-enzymatically from oxidized dopamine or norepinephrine metabolites [24,25]. NM pigments are encapsulated by macroautophagy, forming NM organelles that also contain proteins and lipids [26] and that have a very slow turnover, remaining inside the neuron for the rest of its life [27]. NM levels increase strongly in aging particularly in SN and locus coeruleus (LC), yet NM is present also in other brain regions but at much lower concentrations [28]. In the SN, NM levels increase almost linearly with age [29,30]. LC has a low total Fe content which does not increase considerably with age, however, the LC neuronal NM (and NM stored Fe) levels do, although less strongly than in SN.

After uptake into endothelial cells of the blood–brain barrier (BBB), Fe is further exported to brain interstitial fluid via the ferroportin (Fpn) protein as ferrous ion which gets oxidized to ferric ion by ceruloplasmin and loaded onto transferrin. Ferritin expression is regulated mainly at post-transcriptional level by iron regulatory proteins (IRPs) which bind specific RNA sequences called iron-responsive elements (IREs) in the 5′ untranslated region of ferritin mRNA [31]. Ferric transferrin in the interstitial fluid is taken up by neurons and glial cells by transferrin-receptor (TfR) mediated endocytosis. The ferroportin protein level is controlled by the master Fe regulatory peptide hepcidin which binds to ferroportin (hepcidin-ferroportin binding is 80-fold stronger in the presence of Fe [32]), after which the ferroportin-hepcidin complex is internalized and degraded in lysosomes, thereby stalling cellular Fe from export. As the brain has limited ability to export excessive Fe back to the systemic circulation, BBB has a paramount role in regulating CNS metal levels. Any deviations from a homeostatic state, such as genetic aberrations affecting metal excretion, or enhanced environmental exposure to Fe [4], will allow for Fe accumulation possibly predisposing for future neurodegeneration. It was shown that Fe and other metals accumulate in the choroid plexus under various conditions [33] and Fe is considered a metal of intermediate toxicity in relation to causing damage to the BBB and the blood-cerebrospinal fluid barrier (BCSFB) [33,34]. It is theoretically possible that dysfunction of BBB/BCSFB may predispose to increased transfer of Fe to the brain compartment.

Systemically, a chronic low-grade inflammation (so called inflammaging) develops during aging [35,36] (Figure 1). Simultaneously, the immune system deteriorates. Thus, in normal aging, a mild neuroinflammation can be anticipated, which may be theoretically related to environmental exposures [37,38]. Inflammation can intensify in neurodegenerative diseases, e.g., Alzheimer’s disease (AD) characterized by extracellular plaques (common also in normal aging) that are attacked by iron-rich microglia, which express high levels of nicotinamide adenine dinucleotide phosphate oxidases (NOX) releasing O_2_^•−^, facilitating formation of H_2_O_2_. Furthermore, the severity of neurodegenerative disorders such as amyotrophic lateral sclerosis (ALS), multiple sclerosis (MS), and stroke is positively associated with serum ferritin levels [39]. Increased levels of the proinflammatory marker interleukin-6 (IL-6) may increase phosphorylation of Janus kinase (JAK), upregulate hepcidin, and downregulate ferroportin, which may stall export of Fe as explained above [40,41,42]. The connection between systemic and brain Fe deposition and inflammatory markers in PD was suggested in a recent study [43]. CSF from neurodegenerative dementias such as AD, frontotemporal lobar degeneration, and prion diseases share a neuroinflammatory marker profile characterized by increased levels of chitotriosidase protein1, YKL-40, and glial fibrillary acidic protein, a structural component of the cytoskeleton of astrocytes [44]. Yet, to what extent these markers are related to brain Fe accumulation as well as the role of CSF ferritin in relation to neuroinflammation remain unclear [43].

If a modest systemic inflammation, which invariably involves neuroinflammation, escalates during aging, neuronal Fe export is halted by systemic hepcidin upregulation while neuronal Fe import may continue, leading to accumulation of Fe intracellularly. Protection against Fe entering the brain by increasing brain hepcidin at the BBB was recently suggested as a remedy for disorders with Fe accumulation [1]. A recent study employed a virus-based strategy to overexpress hepcidin in the brain of young rats which prevented dopamine neuronal loss and decreased Fe and alpha-synuclein accumulation in both rotenone and 6-hydroxydopamine Parkinson’s disease (PD) models [45].

At the subcellular level sequestration of Fe in lysosomes increases the expression of proinflammatory cytokines such as tumor necrosis factor α evoking inflammatory signatures in the brain [46]. Iron dyshomeostasis affects neurons and glia alike and contributes to acidification of lysosomes and to mitochondrial dysfunction, where not only Fe deposition but also Fe deficiency at the subcellular level, and an inadequate intracellular Fe distribution may trigger neuroinflammatory processes [47]. It has been suggested that cellular Fe retention may accelerate aging by damaging DNA and blocking of normal genomic repair, a process referred to as ferrosenescence [48].

Iron has a role in physiological age-related cognitive decline. A recent systematic review identified 41 human studies of the relationship between brain Fe in different regions and cognition. The authors concluded that there is consistent evidence that brain Fe is causally related to cognitive impairment [49]. Striatal Fe was found to be highest in the elderly who also show increased neuroinflammation when assessed through myo-inositol, a marker of activated microglia in the brain, measured by magnetic resonance spectroscopy [50]. Another recent study found decreases in memory performance related to increased Fe, measured by MRI R2* relaxometry, and subsequent gliosis in hippocampus and caudate [51]. Interestingly, a genome-wide association study identified several loci in the heme iron metabolism pathway associated with health-span and longevity; multivariate Mendelian randomization of iron-related traits showed a protective effect of transferrin and deleterious effect of serum Fe [52]. To that point, serum ferritin concentration was found to increase with aging [53].

**Table 1 biomolecules-12-00714-t001:** Patterns of iron metabolism disturbances and accumulation in aging and neurodegenerative disorders.

Condition	Whole Body Level	Brain Level		
		Macroscopic	Cellular	Subcellular
Aging	↑ serum ferritin [53]	↑ Fe in red nucleus, putamen, substantia nigra, dentate nucleus, globus pallidus, caudate nucleus, subthalamic nucleus, cortex [7]	Fe remains stable in oligodendroglia; Fe accumulates in astrocytes and dystrophic microglia in cortex and deep gray matter [12]	Fe bound to ferritin in cytoplasm of microglia and astrocytes and to neuromelanin in neurons [12,28]
AceruloplasMinemia	↑ Fe in liver, pancreas, retina ↑ serum ferritin, ↓ transferrin saturation [54,55]	↑ Fe in putamen, caudate, lateral, habenular, and pulvinar thalamic nuclei, red nucleus, dentate nucleus, inner cortical layers, hippocampus, mammillary bodies, superior and inferior colliculi [56,57]	↑ Fe in astrocytes, neurons [58,59,60]	Fe stored in ferritin/ hemosiderin in lysosomal dense bodies and cytoplasmic inclusions [61,62]
Hereditary Ferritinopathy	↑ Fe in liver, kidney, skin, muscle ↓ serum ferritin [63,64,65]	↑ Fe in globus pallidus, substantia nigra, dentate nucleus, putamen, thalamus, caudate, deep cortical layers [66,67,68]	↑ Fe in nuclei and cytoplasm of microglia, oligodendroglia, neurons, and also extracellularly [64,69]	Fe stored in inclusion bodies consisting of abnormal ferritin aggregates [69,70]
Pantothenate Kinase-Associated Neurodegeneration	-	↑ Fe in globus pallidus, substantia nigra [71,72,73,74]	↑ Fe in astrocytes, neurons, perivascular macrophages, iron dust in neuropil [75,76]	Fe stored in cytoplasmic inclusions co-localized with ferritin [75]
Mitochondrial Membrane Protein-Associated Neurodegeneration	-	↑ Fe in globus pallidus, substantia nigra, putamen, caudate [77,78]	↑ Fe in perivascular macrophages, astrocytes, neurons [79,80]	n.a.
Phospholipase A2-Associated Neurodegeneration	-	↑ Fe in globus pallidus, substantia nigra, dentate nucleus [81,82,83]	↑ Fe perivascularly in extracellular deposits and in macrophages [82,84,85]	n.a.
Beta-Propeller Protein-Associated Neurodegeneration	↑ serum Tfr/logFerrit ratio [86]	↑ Fe in substantia nigra, cerebral peduncles, globus pallidus [87]	↑ Fe in excessive macrophages [88,89]	n.a.
Friedreich Ataxia	Fe-positive granules in cardiomyocytes ↓ serum Fe [90,91,92,93]	↑ Fe in dentate nucleus, red nucleus [94,95]	Fe switched from oligodendroglia to microglia in dentate nucleus [96,97,98]	Fe presumably primarily accumulated in mitochondria [99,100]
Wilson Disease	↑ Fe in liver ↑ serum Fe, ferritin, hepcidin, soluble transferrin receptor [101,102,103]	↑ Fe in globus pallidus, putamen, caudate, thalamus, substantia nigra, red nucleus, subthalamic nucleus [104,105]	↑ Fe in excessive macrophages, astrocytes [106]	n.a.
Parkinson Disease	↓ Fe in serum/plasma [107]	↑ Fe in substantia nigra [108,109]	↑ Fe in neurons and adjacent neuropil, microglia, perivascularly in extracellular deposits [110,111,112]	Fe bound to neuromelanin in dopaminergic neurons [112,113]
Alzheimer Disease	↓ Fe in serum/plasma [114,115,116]	↑ Fe in (mostly temporal) cortex, globus pallidus, caudate, putamen [117,118,119,120,121,122]	↑ Fe in amyloid plaques, microglia, along myelinated fibers [117,123,124,125]	Fe bound to amyloid partially composed of magnetite nanoparticles [126,127]
Amyotrophic Lateral Sclerosis	↑ ferritin, ↓ transferrin in serum [128] ↑ Fe in liver, kidneys [129] ↑ Fe in spinal cord [130,131]	↑ Fe in motor cortex, caudate, subthalamic nucleus, globus pallidus, substantia nigra, red nucleus [132,133,134]	↑ Fe in spinal cord neuron nuclei [135] ↑ Fe in microglia in motor cortex [136]	n.a.
Multiple Sclerosis	-	↑ Fe in globus pallidus, putamen, caudate ↓ Fe in normal appearing white matter and thalamus [137,138,139,140,141]	↑ Fe in macrophages, activated microglia in the rim of lesions; in reactive astrocytes in the inactive centers of lesions; in oligodendroglia, astrocytes, and microglia in the deep gray matter [137,142,143,144,145]	Fe in active lesions stored in ferritin, hemosiderin, and magnetite [144]

↑, increased; ↓, decreased; Fe, iron; TfR, transferrin receptor.

## 3. Neurodegenerations with Brain Iron Accumulation (NBIA) Group

Genetic disorders caused by pathogenic variants in genes coding for proteins directly involved in the Fe regulatory pathway, aceruloplasminemia (Online Mendelian Inheritance in Man (OMIM) database entry #604290) and hereditary ferritinopathy (OMIM #606159) are of great interest since their study may bring important information regarding consequences of cerebral Fe dyshomeostasis, which may ease the understanding of the role of Fe in more common sporadic neurodegenerative disorders.

***Aceruloplasminemia***, caused by pathogenic variants in the ceruloplasmin gene compromising its ferroxidase activity and preventing cellular Fe efflux, is characterized by adult-onset diabetes, retinopathy, hepatopathy, and neuropsychiatric symptoms. Clinical manifestations are presumably a consequence of multiorgan damage due to systemic Fe accumulation affecting liver, pancreas, retina, and many brain regions [146]. The molecular mechanism of Fe accumulation is not completely understood, particularly the aspect why in human patients and a rat animal model Fe accumulates in cells such as hepatocytes and astrocytes but not in macrophages [147]. Abnormal Fe metabolism, i.e., microcytic anemia with low transferrin saturation, high serum ferritin, and profound brain Fe accumulation can be detected already in asymptomatic subjects, often in childhood [148]. The regional distribution of brain Fe deposits resembles that observed in normal aging with putamen, caudate, lateral and pulvinar thalamic nuclei, red nucleus, dentate nucleus, and inner cortical layers in frontal and occipital lobes being the most affected regions (Table 1). Additionally, Fe accumulation also affects the lateral habenula, mammillary bodies, superior and inferior colliculi, and hippocampus [56]. Ex-vivo examination of brain tissue using quantitative MRI, Electron Paramagnetic Resonance (EPR), and Superconducting Quantum Interference Device (SQUID) magnetometry showed that more than 90% of accumulated Fe is deposited as ferrihydrite-Fe in ferritin and hemosiderin which are the main drivers of increased R2* relaxivity and susceptibility; the remaining part of Fe was found to be embedded in oxidized magnetite/maghemite minerals with ferrimagnetic properties [61]. Iron accumulation in putamen may rarely be accompanied with cavitation [149]. Another ex-vivo study employing X-ray microanalysis suggested that substantial Cu deposits may coexist with lysosomal hemosiderin-Fe deposits. However, sources of these Cu deposits remain unknown [150]. Based on a literature review, it was suggested that chelation treatment in neurologically symptomatic patients rarely leads to improvement while a long-term treatment in neurologically asymptomatic patients may prevent the occurrence of neuropsychiatric symptoms [151]. Others have documented improvement of neurological symptoms in 50% of patients with neurological symptoms [152]. An aggregated case study analyzed chelation treatment outcomes in 24 symptomatic and 24 asymptomatic patients and concluded that clinical improvement was described in almost half of the symptomatic patients while chelation postponed symptom onset by approximately 10 years in asymptomatic patients [153]. Animal studies have shown that ceruloplasmin replacement may decrease brain Fe burden, particularly in the choroid plexus, and lead to amelioration of neurologic symptoms [154,155]. Combination of Fe chelation and treatment with fresh frozen plasma, believed to restore ceruloplasmin levels, reportedly improved neurologic symptoms in a middle-aged aceruloplasminemia patient [156]. Altered functions of ceruloplasmin ferroxidase activity were suggested to be connected to several neurodegenerative disorders, hence it is interesting to investigate neurologic status in heterozygous carriers of ceruloplasmin variants [157]. A systematic review of reported heterozygous cases confirmed slightly decreased serum ceruloplasmin levels that were associated with brain Fe accumulation and neurologic abnormalities in 7 out of 21 cases, ataxia and cerebellar atrophy being the most common [158]. Neurologic symptoms in these patients were observed despite normal serum Fe and ferritin levels suggesting that the brain may be more susceptible to mild changes in ceruloplasmin activity compared to other organs.

***Hereditary ferritinopathy*** is an autosomal dominant disorder caused by pathogenic variants in the *ferritin light chain (FTL)* gene (recently reviewed in [159,160,161]). Variants in exon 4 affecting the C-terminal residues of the FTL protein are associated with a neurological disorder also referred to as neuroferritinopathy. These variants were shown to cause changes in amino acid sequence creating subunits that assemble into ferritin polymers with altered structure and/or much larger pores than the wild type. Polymers containing mutated protein presumably have lower stability and reduced Fe retention ability leading to an increased cytosolic labile Fe pool [159,162,163]. Neuroferritinopathy is manifested by low serum ferritin along with pathological Fe deposits localized in various brain regions, most consistently in the globus pallidus, SN, and dentate nucleus, but also in the putamen, caudate, and deeper cortical layers (Table 1) [164]. Deep gray matter structures, mostly globus pallidus, may be affected by necrosis and it was suggested that the destructive process may begin in the medial medullary lamina in globus pallidus [69]. Thus, the pattern of Fe accumulation and likely also neuropathological mechanisms are different from those observed in aceruloplasminemia. Indeed, a neuropathological study examining six neuroferritinopathy brains found increased Fe concentration only in cortical regions and thalamus while Fe content in the globus pallidus, putamen, and cerebellum was not significantly increased compared to controls. On a microscopic level, tissue Fe distribution was altered with deposits detectable in nuclei and cytoplasm of cells with oligodendrocyte morphology, in microglia and neurons, whereby Fe was frequently bound to inclusion bodies consisting of abnormal FTL aggregates. The latter study found no evidence of oxidative stress assessed by quantitative analysis of heme oxygenase-1, protein carbonyl groups, superoxide dismutase (SOD) 1 and 2, and heat-shock protein 70 expression in brain tissue [69]. Overall, results of this study suggest that abnormal FTL protein overexpression, aggregation, and consequent proteostasis may be the primary cause of neurodegeneration while derangement of Fe metabolism may occur as a secondary event. Theoretically, decreased T2/T2* signal in the gray matter could be caused by alteration of magnetostatic properties of aberrant Fe-FTL aggregates rather than by Fe accumulation per se; however, the effect of deranged Fe storage in aberrant FTL on its magnetostatic properties were not examined. These findings partially contradict the increase of cytosolic free Fe and oxidative stress found in a previous study employing cellular FTL model [165] and may explain the lack of response to chelation therapy in neuroferritinopathy patients. In a mouse model, normalization of systemic Fe metabolism was observed while brain pathology was not influenced by chelation treatment [166]. Despite the fact that the etiopathogenetic involvement of Fe in FTL is unclear, human neuropathological studies are in line with research employing cellular and animal FTL models in that substantial accumulation of aberrant ferritin is an important factor in the development of neurological dysfunction [160].

Other disorders in the NBIA group are caused by variants in genes not directly related to Fe metabolism. While the number of causative genes is increasing, major NBIA disorders comprise pantothenate kinase-associated neurodegeneration (PKAN) caused by mutation in the *PANK2* gene, mitochondrial membrane protein-associated neurodegeneration (MPAN) caused by mutation in the *C19orf12* gene, beta-propeller protein-associated neurodegeneration (BPAN) due to WDR45 mutation, and phospholipase A2 type G6 (PLA2G6)-associated neurodegeneration (PLAN) due to PLA2G6 mutations (reviewed in [2,167,168,169,170,171,172]).

***PKAN*** (OMIM #234200) manifests with dystonia, parkinsonism, dysarthria, spasticity, psychiatric symptoms, and retinopathy. The PANK2 gene encodes pantothenate-kinase 2, an enzyme involved in coenzyme A synthesis. The brain MRI in PKAN is conspicuous by a central T2 hyperintensity surrounded by T2 hypointense signal representing Fe deposits with central necrosis in the globus pallidus, referred to as eye-of-the-tiger sign [71,72,73,74]. It was shown that streaking T2 hyperintensity alongside the medial rim of globus pallidus predates occurrence of Fe deposits [71] and that Fe accumulation may be even absent in a patient with severe symptoms [173]. Microscopically, iron deposits were detected in astrocytes, perivascular macrophages, neurons, and extracellularly as “iron dust” (Table 1) [75,76]. Some point mutations are not entirely detrimental to the PANK2 protein, leaving residual enzymatic activity that can be assessed in erythrocytes. Patients carrying such mutations tend to have milder disease severity compared to patients with two null mutations resulting in no enzymatic activity which confirms that pathogenesis of PKAN is related to the damage of PANK2 activity [174,175]. There are inconsistent results regarding Fe homeostasis in studies using cellular models. In the study by Arber et al. (2017), neuronal cells derived from Induced Pluripotent Stem Cells (iPSCs) did not show any signs of Fe accumulation and Fe chelation paradoxically led to deterioration of mitochondrial function [176]. In another cellular model using patient-derived iPSCs differentiated into glutamatergic neurons, alteration of calcium homeostasis with calcium phosphate aggregates in the mitochondrial compartment was shown in addition to two-fold increased cellular Fe levels [177,178]. Moreover, globus pallidus calcifications located in the T2 hyperintense region can be detected also at a macroscopic level by brain CT in almost 50% of the patients [177,179,180,181]. PKAN-patient-derived fibroblasts under high environmental Fe condition eagerly accumulated Fe in comparison to control cells. Further experiments suggested that this may be a consequence of impaired TfR downregulation and recycling due to defective palmitoylation [182]. This defect in TfR recycling was by the same group found to be a universal abnormality in cellular models of other NBIAs including MPAN, PLAN, and FRDA [182,183]. Chelation treatment with deferiprone lasting 18 months was able to remove excess Fe from the globus pallidus in a large placebo-controlled trial with 88 patients. Compared to placebo, deferiprone slowed down disease progression only in patients with later age at onset and milder clinical severity. Further evidence of deferiprone efficacy was brought by an extension phase which indicated that the rate of progression in patients switched from placebo to active drug dropped to the values observed in patients treated by deferiprone from the beginning [184]. The latter study indicated that Fe chelation does not lead to clinical improvement and may only work as a preventive and potentially neuroprotective measure in PKAN. Else, the change in evolution of clinical severity under Fe chelation may only become apparent during long-lasting treatment. Supplementation of coenzyme A or, in the case of partially preserved PANK2 activity, pantothenate, rescued the phenotype in cellular and animal models of PKAN [178,185,186]. Iron accumulation was also reduced after pantothenate supplementation suggesting that coenzyme A depletion is directly associated with Fe accumulation [187]. The role of coenzyme A deficiency was further supported by showing that patients with biallelic pathogenic variants in Coenzyme A synthase have phenotype and MRI abnormalities like PKAN [188] and that zebrafish models of coenzyme A synthase and PANK2 deficiency exhibit similar phenotypes [189]. Notably, additional experimental therapeutic strategies based on substrate supplementation were examined in human patients with variable success (reviewed in [190,191].

***MPAN*** (OMIM #614298) typically manifests with progressive spastic paraparesis, psychiatric disturbances, and slow cognitive decline, variably accompanied by motor neuropathy, dystonia, parkinsonism, and visual symptoms [192]. Abnormal Fe deposits are consistently located in globus pallidus and SN while late-onset cases show Fe accumulation also in the putamen and caudate [78], which has been confirmed also in younger MPAN patients by R2* mapping at 7T MRI (Table 1) [77]. Microscopically, iron deposits are found in perivascular macrophages, astrocytes, and neurons [79,80]. Although it was initially described as an autosomal recessive disorder, several patients with only monoallelic pathogenic variants were reported [193,194,195,196,197]. Additionally, increased Fe content in the putamen and caudate nucleus was shown in healthy heterozygous carriers of pathogenic variants suggesting that increased Fe concentration may be an endophenotypic marker of genetic heterozygosity [77]. MPAN patients may develop white matter lesions in later stages [198] while significantly increased glutamate content was found in the subcortical Rolandic white matter using MR spectroscopy [77]. Both findings converge on the conclusion that myelin damage is associated with MPAN, consistent with the presumed role of C19orf12 protein in lipid metabolism and mitophagy [199]. MPAN cellular models exhibited mitochondrial dysfunction, iron overload, and increased oxidative damage that were preventable by antioxidants and iron chelator treatment, a pattern consistent with ferroptosis [200]. The disorder is remarkable in the widespread alpha-synuclein staining and Lewy bodies found on neuropathological examination [195]. Therefore, MPAN research can bring important information regarding the relationship between cerebral Fe accumulation and alpha-synuclein aggregation. Interestingly, decreased heart rate variability and resting tachycardia without heart rate slowing during nighttime was described in MPAN suggesting cardiac autonomic dysfunction [201]. Yet, it was not investigated if abnormal alpha-synuclein aggregates are present in peripheral autonomic nerves as is the case in sporadic synucleinopathies such as PD. Despite promising findings in cellular models [200], no change in clinical status after chelation therapy with deferiprone lasting more than one year was reported in a case study [202].

***PLAN*** (OMIM #610217) phenotypic spectrum includes infantile neuronal dystrophy, juvenile atypical neuronal dystrophy, and adult-onset dystonia-parkinsonism. Infantile neuronal dystrophy is characterized by progressive psychomotor deterioration with spastic or hypotonic tetraparesis, cerebellar ataxia, optic atrophy, and seizures. Patients with later disease onset manifest with pyramidal signs, motor neuropathy, eye movement abnormalities, cognitive decline, psychiatric symptoms, dystonia, and parkinsonism (reviewed in [203,204,205]). PLA2G6 encodes calcium-independent phospholipase A2 group 6 (iPLA2) which hydrolyzes membrane phospholipids, releasing free fatty acids and 2-lysophospholipids with a putative role in phospholipid homeostasis, membrane remodeling, immune responses, and apoptosis. The late onset of symptoms is likely explained by partial preservation of enzymatic activity and in most patients is not associated with Fe accumulation [206,207]. In patients with early onset, Fe accumulation in the globus pallidus and SN is observed in approximately 50% of cases [81,82,83] while progressive cerebellar atrophy and claval hypertrophy are constant findings [208]. Neuropathological findings in PLAN include perivascular iron deposits located extracellularly and in macrophages (Table 1), and widespread neuroaxonal spheroids composed of accumulated membranes with tubulovesicular structures and Lewy bodies which makes this disorder an interesting model for the study of sporadic synucleinopathies [84,85,209,210]. Increased alpha-synuclein expression appears to be a direct consequence of iPLA2 deficiency [211]. Yet, several studies examined the association between PLA2G6 genetic variants and PD arriving at the conclusion that monoallelic PLA2G6 variants do not increase PD risk [212,213,214,215]. Patient-derived iPCs with biallelic PLA2G6 mutation differentiated into dopaminergic midbrain neurons exhibited signs of increased endoplasmic reticulum stress. Azoramide, a modulator of unfolded protein response, prevented the activation of mitochondrial apoptosis pathways through the reduction of endoplasmic reticulum protein folding, calcium dyshomeostasis, ROS production, mitochondrial dysfunction, and caspase 3 activation [216]. In another study using PLAN cellular model based on iPCs, ROS production, lipid peroxidation and mitochondrial dysfunction was attenuated by vitamin E treatment [217]. Interestingly, PLA2G6 was shown to be involved in ferroptosis regulation. Through degradation of hydroxy-peroxy-phosphatidylethanolamine, a compound which triggers ferroptotic pathways, PLA2G6 attenuated ferroptotic signaling in placental trophoblasts [218,219]. To what extent PLA2G6 dysfunction is connected to cerebral Fe accumulation through ferroptosis remains to be elucidated. Mouse and zebrafish models of PLAN confirmed SN degeneration, Lewy body pathology, mitochondrial dysfunction, altered mitophagy, increased ROS production, and endoplasmic reticulum stress [220]. In a drosophila model, lack of iPLA2 activity resulted in shortening of acyl chains in membrane phospholipids likely affecting membrane fluidity with subsequent synaptic dysfunction, endoplasmic reticulum stress, acceleration of alpha-synuclein fibril formation, and degeneration of dopaminergic neurons. Interestingly, the aberrant phenotype was rescued not only by iPLA2 but also by dietary supplements with long-chain free fatty acids and by overexpression of C19orf12, the protein involved in MPAN [221]. This finding suggests that iPLA2 and C19orf12 act in the same lipid pathway defects and their dysfunction may cause Fe accumulation and alpha-synuclein aggregation through similar mechanisms. In other studies employing PLAN drosophila models, loss of neuronal function, loss of psychomotor activity, impaired retromer function, and lysosomal accumulation of ceramide with expansion of lysosomes were found. Similar phenotype as with iPLA2 dysfunction was also observed with alpha-synuclein overexpression [222,223].

***BPAN*** (OMIM #300894) manifests as global developmental delay with epilepsy and Rett-like syndrome in childhood and rapid-onset dystonia-parkinsonism and progressive dementia in early adulthood (recently reviewed in [224,225,226]). The WD-domain repeat 45 (WDR45) gene encode a beta-propeller WD repeat protein interacting with phosphoinositides (WIPI4), which is critically involved in the autophagosome-lysosome fusion in neuronal cells [227]. It was suggested that WDR45 has a role in fusion between adjacent membranes in general as alterations in endoplasmic reticulum and mitochondrial morphology were shown in cellular and animal BPAN models [228]. Infantile onset of clinical symptoms precedes the occurrence of Fe deposits visible on brain MRI. It was shown that the earliest imaging abnormalities may be dentate nucleus and SN swelling visible as T2 hyperintensity accompanied by corpus callosum thinning, cerebral atrophy, and delayed myelination [229]. Transient T2 hyperintensities were also described in globus pallidus and SN in infants following episodes of febrile seizures suggesting they may reflect the primary neuropathological damage. Iron deposits are first apparent at 2–7 years of age with iron-sensitive MRI techniques and increase progressively over time [87]. Iron accumulation is most pronounced in the SN extending into cerebral peduncles and consistently present in the globus pallidus [230]. Patients with globus pallidus calcifications were also reported [231]. On a microscopic level, iron deposits are mostly present in macrophages (Table 1) [88,89]. Several studies have found connections between WDR45 mutations and Fe metabolism. Iron overload, accompanied by reduced cellular ferritin levels and increased markers of oxidative stress and mitochondrial damage, was observed in patient-derived fibroblasts and dopaminergic neurons [232]. Another study employing patient-derived fibroblasts found upregulation of DMT1 and downregulation of TfR associated with cellular Fe accumulation induced by nutrient starvation [233]. This pattern of molecular abnormalities led to the hypothesis that mutations in WDR45 may affect ferritinophagy and mitophagy causing deranged Fe recycling with DMT1 upregulation as a proximal cause of Fe accumulation [224]. Impaired ferritinophagy and non-transferrin-bound iron pathway were confirmed to mediate the accumulation of iron in another cellular BPAN model [234] whereby iron was detected mostly in lysosomal vesicles [235]. Yet another study employing the cellular BPAN model found accumulation of TfR in cells overexpressing mutated WDR45 that was associated with increased Fe levels [236]. Interestingly, increased ratio between serum concentration of soluble TfR and logarithm of serum ferritin concentration (sTfR/logFerrit) was observed in five BPAN patients, suggesting complex systemic disruption of Fe metabolism [86]. Chelation therapy was attempted in two BPAN patients. Deferiprone treatment with the doses 250 and 1000 mg twice a day for several months did not lead to any clinical improvement, instead, worsening of neurologic symptoms were reported [237,238].

Other rare genetic disorders were suggested to belong to the NBIA group, mostly based on brain MRI findings, i.e., decreased basal ganglia signal on T2/T2* or susceptibility weighted images (reviewed in [239]): Fatty acid 2 hydroxylase-associated neurodegeneration [240], Woodhouse–Sakati syndrome caused by mutations in *DCAF17* gene [241,242], Kufor–Rakeb syndrome caused by *ATP13A2* mutations [243,244], Adaptor protein complex 4 (AP4) deficiency [245], sterol carrier protein 2 (SCP2) deficiency [246], DDHD1 encoding phospholipase A1 [247], FBXO7 deficiency [248], GTP-binding protein 2 (GTPBP2) deficiency [249,250], fucosidosis [251], GM1 gangliosidosis [252], VAC14 syndrome [253], and *TBCE* [254], *KMT2B* [255], *IRF2BPL* [256], *REPS1*, and *CRAT* mutations [182]. Further research into brain Fe homeostasis in these disorders may generate important information regarding mechanisms and consequences of brain Fe disturbances. However, only small numbers of patients have been described and/or Fe accumulation is inconsistent, and neuropathological proof of Fe accumulation is lacking in these disorders [2]. Caution is necessary in the interpretation of T2/T2*-weighted MRIs as has been shown for mannosidosis; this disorder was considered in the differential diagnosis of NBIA [257] but a semiquantitative analysis in a group of mannosidosis patients showed that the hyposignal appearance of the deep gray matter is only relative and rather caused by an increased signal from the surrounding white matter [258].

## 4. Other Genetic Disorders with Brain Iron Deposits

***Friedreich’s ataxia*** (FRDA; OMIM #229300) is an autosomal recessive disorder caused by frataxin mutations. The degeneration in FRDA first affects dorsal root ganglia and posterior spinal columns, typically during childhood or early adulthood, followed by pathology in corticospinal and spinocerebellar tracts, cerebellar dentate nuclei, and to a lesser extent also in cerebral tissue [259]. Abnormal Fe deposits in FRDA have been detected in the cerebellar dentate nuclei by MRI by some groups while others could not confirm this finding [260]. The most recent MRI studies employing QSM found increased Fe concentration in dentate and red nuclei (Table 1). Moreover, Fe content in the dentate nuclei increased upon longitudinal follow-up and correlated with disease severity [94,95,98]. Intriguingly, despite findings of abnormal Fe distribution and altered expression of molecules involved in Fe trafficking, no conclusive evidence of bulk Fe accumulation in dentate nucleus, dorsal root ganglia, or cardiomyocytes has been shown by ex vivo analyses [92,261]. On the other hand, plasma Fe was found to be decreased in FRDA compared to controls. This may be theoretically explained by upregulation of intracellular Fe uptake that leads to extracellular Fe deficiency [93].

In FRDA, Fe accumulates in mitochondria, supposedly because of decreased mitochondrial capacity to utilize Fe for the synthesis of Fe-sulphur (Fe-S) clusters. It was hypothesized that defective Fe-S cluster synthesis leads to upregulation of mitochondrial Fe uptake [261]. The essential role of frataxin in mitochondrial Fe metabolism is only partly understood, with most recent evidence suggesting a role for frataxin as an activator of cysteine desulfurase, which is a key enzyme in the synthesis of Fe-S clusters [261]. Consequently, Fe-S containing proteins, such as aconitase and respiratory complexes I–III are deficient, resulting in impaired energy metabolism, oxidative damage, and progressive mitochondrial dysfunction. Increasing evidence indicates the presence of ferroptosis markers in FRDA, including increased plasmatic malondialdehyde documenting lipid peroxidation, low levels of glutathione and downregulation of GPX4 implicating lower antioxidant status, upregulation of nuclear receptor coactivator 4 (NCOA4) indicating ferritinophagy, and downregulation of NRF2, which is a natural inhibitor of ferroptosis [262]. Along the same line of thinking, skin fibroblasts from FRDA patients derived from biopsied tissue exhibit higher phototoxicity to UVA radiation caused by an increased mitochondrial labile Fe pool and ROS as compared to healthy controls. Interestingly, pretreatment with a mitochondrial Fe chelator could prevent ROS production and rescue fibroblasts from UVA-induced cell death [99]. Ferroptosis inhibitors are currently studied in FRDA disease models [263]. In addition to Fe accumulation, studies in cellular disease models and human post-mortem tissue have pointed towards disturbed calcium homeostasis in FRDA. Increased cytoplasmic calcium levels related to frataxin dysfunction were found to be associated with accelerated apoptotic cell death in cultured neuronal cells [264].

The efficacy of Fe chelation therapy using deferiprone was examined in several clinical studies which indicated a possible improvement or stabilization of neurologic and cardiac function with low doses, particularly in patients with short disease duration, while higher doses led to clinical worsening (reviewed in [151]). In an observational study of 5 FRDA patients treated with low dose deferiprone and indebenone for 10–24 months, neurologic improvement was observed only in one patient and improvement of cardiac function in another one patient. These observations highlight the inconsistent results of Fe chelation treatment in FRDA [265].

***Wilson’s disease*** (WD) is a hereditary systemic Cu toxicosis resulting from mutations in the gene coding for ATP7B which is responsible for Cu transport and efflux from liver cells. Neurologic symptoms are consequent to cerebral Cu accumulation; Cu levels in the WD brain are diffusely increased by the factor of ten (reviewed in [266,267]). It becomes increasingly clear that Fe metabolism is also impaired in WD. Serum Fe, ferritin, hepcidin, and soluble TfR were higher while serum transferrin was lower in treatment naïve WD patients compared to controls. Anti-Cu treatment partially normalized Fe parameters but differences in comparison to controls were still noted. These changes in Fe homeostasis are likely a result of a systemic inflammatory response that is attenuated with long-term anti-Cu treatment [101]. It was suggested that alterations in Fe metabolism are more pronounced in males than in females and may underlie more severe neuropsychiatric symptoms in male WD patients [268]. Systemic inflammatory response with increased plasma prostaglandins and other oxylipins was found in WD [269]. Additionally, increased serum IL-1 and TNF-alpha was observed in patients with the most severe brain MRI abnormalities [270]. Increased Fe concentration was observed also in the liver, particularly in hepatocytes [102,103,271,272]. Serum ferritin was found to be an independent predictor of liver injury in WD, similarly to other liver disorders [273].

Hypointense signal in the basal ganglia is consistently observed on T2/T2* weighted MRIs. A post-mortem MRI-histopathology correlation study showed that R2* relaxometry in the putamen, caudate nucleus, and globus pallidus is related to Fe concentration and number of Fe positive macrophages but not to Cu content [106]. Additionally, density of Fe-positive macrophages was associated with neuropathological severity [106]. In long-term treated neurologic WD patients, increased Fe concentration on brain MRI was observed diffusely in deep gray matter unrelated to clinical severity [104,105]. In another study which correlated MRI findings and clinical severity in WD, Fe deposits were positively associated with severity of neurological symptoms and the degree of atrophy in the caudate nucleus, globus pallidus, putamen, dentate nucleus, and thalamus [270]. Disparate results may be due to different inclusion criteria, the former study included only stable patients on a chronic anti-Cu treatment while the latter study also included patients in the active phase of the disease shortly after treatment initiation. Accordingly, in a WD case with severe paradoxical worsening on therapy, rapid cerebral Fe accumulation was observed after initiation of chelation treatment, and it was accompanied by profound tissue loss [274]. It is thus possible that Fe deposits may be related to clinical severity in the early phase of treatment while other factors may become more important in the chronic stage. The mutation p.C282Y in the homeostatic Fe regulator protein (HFE) gene is the most common cause of hereditary hemochromatosis and it was shown that it is associated with an earlier age of onset in WD [275]. Interestingly, a similar finding was previously reported in FRDA [276].

Impaired Cu incorporation into ceruloplasmin leading to low ferroxidase activity has been hypothesized to underlie Fe accumulation in WD [277]. This notion was supported by findings in a WD mouse model where mild anemia, decreased plasma Fe levels together with Fe accumulation in hepatocytes, and liver macrophages were found [278]. However, abnormalities observed in the latter model are consistent with findings in aceruloplasminemia but do not mirror the situation in human WD patients.

## 5. Iron Accumulation and Pathology in Sporadic Neurodegenerative Disorders

Neurodegenerative diseases such as AD, PD, and ALS are sporadic in most cases and exhibit Fe homeostasis perturbations [279,280]. Environmental factors such as exposure to foreign chemicals and metals (xenobiotics), nutrition, and lifestyle in combination with genetic susceptibility likely affect disease onset and progression. Accumulating evidence indicates a role for Fe deposits in these conditions. In humans, neurons of certain regions such as SN and LC build-up NM which successively sequesters excess labile Fe. Interestingly, these regions are often affected early in neurodegenerative diseases, SN in PD, and LC in AD and PD [281,282,283,284]. Protein aggregates (fibrils), hallmark of neurodegenerative disorders, seem to attract Fe and other metals which act as a glue and contribute to protein misfolding. As the aggregates are not removed, they accumulate various adducts and become increasingly damaged by reactive aldehydes, quinones, and ROS. Interestingly, the proteins overexpressed in affected neurons—amyloid precursor protein (APP) in AD and alpha-synuclein in PD—both contain atypical (type II) Fe responsive IREs in the 5′ untranslated region (UTR) of their mRNA [285,286]. Thus, increased cytosolic Fe may trigger translation of APP and alpha-synuclein. Iron levels are 4- to 5-fold higher and formation of NM with age is stronger in SN than in LC [287]. Since NM formation also binds Cu [287,288], present at a very low free concentration in the cytosol, NM formation may indirectly cause Cu deficiency in catecholaminergic neurons. Neuronal cytosolic Cu levels may also be lowered by continuous export of APP, which holds a Cu binding domain, through trans-sarcolemmal passage where APP is cleaved into peptides [38,285]. Notably, with disease progression, the neurons richest in Fe or most deprived of Cu die [285] and results from metal analysis can therefore be misleading depending on disease stage.

**PD** is the second most common neurodegenerative disease after AD. Age, certain chemicals (e.g., 1-methyl-4-phenyl-1,2,3,6-tetrahydropyridine (MPTP), paraquat, chlorpyrifos, rotenone, and maneb), and metals are risk factors for sporadic PD [37,38,289,290,291,292]. Pathological effects include alpha-synuclein aggregation inside Lewy bodies in susceptible neuronal populations, dopaminergic neuronal Fe accumulation in SN, oxidative stress, and SN neuronal loss. Alpha-synuclein is a neuronal synaptic protein that regulates synaptic vesicle trafficking and subsequent neurotransmitter release. Moreover, SOD1 aggregates, more often seen in ALS, can be present [293]. Brain regions affected early in PD include SN, LC, olfactory bulb, and dorsal motor neurons of the vagus nuclei [281,282]. However, iron accumulation in PD is limited to SN pars compacta [108,109].

A hypothesis of how PD-inducing xenobiotics (exemplified by MPP^+^, a MPTP metabolite having NM affinity [290]) may exert toxicity is illustrated in Figure 2. NM, whose levels increase in SN with age, attracts MPP^+^ that can diffuse into mitochondria and inhibit the respiratory chain, causing decreased adenosine 5′-triphosphate (ATP) and increased ROS production. MPP^+^ (structurally very similar to paraquat) may also redox-cycle in the cytosol (produces ROS) or could potentially directly oxidize DA. Enhanced ROS levels, together with increased catalytic labile Fe in aging, mediates oxidation of DA into reactive DA metabolites such as DA-ortho-semiquinone radical (DA-*o*-SQ^•^) and 6-hydroxydopamine that both can further oxidize into quinones. Misfolded and aggregated proteins in the cytosol (e.g., alpha-synuclein) are attacked e.g., by reactive DA-*o*-SQ^•^, DA-quinone, or aldehydes (e.g., 3,4-dihydroxyphenyl acetaldehyde (DOPAL) and/or aldehydes generated during lipid peroxidation) that form toxic protein adducts which may serve as starting seeds for the molecular backbone in the NM synthesis [37]. The natural NM backbone also contains cysteinyldopamine. Eventually, multiple NM pigments get stored away inside tiny NM organelles inside the cytosol [37]. Aged SN neurons appear particularly vulnerable to toxic insults due to the presence of higher labile Fe levels, which in combination with ROS-producing xenobiotics (attracted to NM) and/or ROS from inflammation, oxidize DA into reactive DA metabolites. Labile Fe and ROS may also initiate lipid peroxidation, a process generating reactive aldehydes that could ultimately trigger ferroptosis. Eventually, stressed dying catecholaminergic neurons release components such as NM that are attacked and disintegrated by microglia producing ROS and inflammatory cytokines as well as releasing NM organelle stored toxic xenobiotics (metals and chemicals), all of which contribute to neurodegeneration [27,37].

Some support for the hypothesis that Fe accumulation in the brain causes disease can be found from a study in anemic Korean older adults, who showed lower risk of developing PD [294]. On the other hand, a meta-analysis showed mildly decreased serum Fe levels in PD patients compared to controls [107]. A recent study found that in the SN of elderly individuals, oligodendroglial and astroglial cells contained the highest cellular Fe concentrations, whereas in PD, the Fe concentrations were increased in most cell types, including neurons in the SN [112]. In contrast, PD patients show decreased Cu levels in neurons containing NM [295]. Another study observed increased Fe and inflammatory marker IL-1 beta in CSF in PD, particularly in PD patients with excessive daytime sleepiness [296]. Furthermore, increased serum levels of hepcidin and IL-6 in PD patients were described [297]. Long-term oral administration of Fe-citrate to 9 months old mice caused parkinsonism, particularly at higher doses [298].

In efforts to treat elevated brain Fe concentrations, a conservative low dose iron chelation mode is preferred. The aim of such chelation is to clear local siderosis from aberrant labile Fe pools, thus redeploying it to physiological cell acceptors or to transferrin without interfering with essential local functions or with hematological parameters [299]. Deferiprone, an Fe chelator, given to PD patients in a dose of 30 mg/kg/day, slightly improved motor symptoms and decreased SN Fe content after 12 months of treatment [300].

**AD** is the most common form of dementia, which is a leading cause of disability in old age. The cardinal neuropathologic features of AD include extracellular accumulation of amyloid-beta amino acids 1–42 (Aβ1-42) peptides and intracellular deposits of hyper-phosphorylated tau (p-tau) in neurofibrillary tangles (NFTs). In CSF, decreased Aβ1-42 together with increased total-tau or p-tau protein levels is found [301]. In AD, p-tau is observed early in the LC, from which it successively spreads to the entorhinal cortex, hippocampus, and eventually to the frontal cortex [302]. The pathogenesis of AD is commonly linked to death of cholinergic neurons and a deficiency in acetylcholine.

Cerebral Fe accumulation in AD was already described in the early 1950s [303]. Lately, Fe accumulation and dyshomeostasis in AD have attracted increasing attention as part of the pathogenesis and as a therapeutic target in recent reviews [304,305,306]. More recent studies of AD brains with advanced techniques (i.a., laser ablation-inductively coupled plasma-mass spectrometry) have shown accumulation of Fe in the frontal cortex and hippocampus apparently together with Aβ and p-tau deposits [117,307,308,309,310,311,312]. Iron accumulation was also found in the inferior temporal cortex but only in AD cases diagnosed clinically and histopathologically [118]. MRI also showed elevated levels of Fe in the brain of AD patients, namely in the hippocampus, various cortical regions, putamen, caudate nucleus, and globus pallidus [119,120,121,306]. The ability of MRI to detect increased Fe levels in AD has been validated by post-mortem MRI and histopathology [117,312]. Interestingly, cortical Fe deposition was found also in other dementias and its location may aid in neuropathological diagnosis; in frontotemporal lobar degeneration, tau pathology was associated with Fe deposits in activated microglia in deep cortical layers, while in TDP pathology, Fe deposits were found in perivascular astrocytic processes in upper cortical layers [279].

Development of Aβ plaques is a common phenomenon in most people in aging. Aβ plaques bind metals including Fe, Cu, and Zn. However, not everyone with Aβ plaques shows cognitive decline. Based on postmortem analyses, up to 44% of all individuals display Aβ deposits with no clinical signs, which may be due to a “cognitive reserve”. However, individuals with both increased Aβ plaques and severe Fe deposits are highly likely to develop dementia, possibly implicating that a tendency towards brain Fe accumulation during aging could be a factor increasing AD susceptibility. In ex vivo brain tissue with AD pathology, cortical Fe in the inferior temporal cortex was strongly associated with accelerated cognitive decline and weakly associated with the extent of proteinopathy [118]. Iron accumulation and increased ferritin expression was identified in microglial cells that cluster around Aβ plaques in AD [123]. AD brains were found to have more of Fe in its more reactive ferrous state than controls that apparently mainly contained ferric Fe [313]. Using MRI and tau-PET, a relationship between Fe accumulation and tau accumulation in AD was seen [120].

A key question is to what extent Fe accumulation in AD takes place secondary to extracellular plaque formation and intracellular NFT accumulation or is an independent phenomenon. While the amyloid hypothesis postulates that Aβ toxicity is the primary cause of neuronal and synaptic loss, there is increasing evidence in the literature for brain metal dysregulation and oxidative damage to neurons in AD; and that these processes start early and escalate throughout the disease. Using synchrotron X-ray spectro-microscopy it was found that amyloid plaque cores isolated from AD brains contain highly reactive nanoscale elemental Cu and Fe (Cu^0^ and Fe^0^) different from their oxide counterparts [314] as well as other reduced forms of Fe [126]. The possible role of these reactive nanoscale metal species in the pathogenesis of neurodegenerative diseases is not known.

Further evidence of Fe dysregulation in AD is that high CSF ferritin levels are related to the transition of mild cognitive impairment (MCI) into AD [315]. In carriers of the ApoE4 allele, a risk factor for AD, cognitive decline was related to increased CSF ferritin levels [316]. An emerging hypothesis suggests that dysregulation in cerebral Fe may contribute to NFT formation as binding of Fe to p-tau proteins appear to precede NFT formation [317]. Iron is speculated to induce conformational changes of the protein. Aβ plaques have been shown to bind various metals that can glue the plaque together as well as form ROS, hence, accumulation of Fe in plaques seems to promote their progression [318]. The Aβ plaque can be subject to removal attempts by microglial cells with NOX facilitated release of O_2_^•−^ and other ROS.

Accumulated Fe can promote oxidative damage of surrounding molecules like proteins, lipids, and nucleic acids. Neuroinflammation, disruption of membrane integrity [319], and reduced Aβ clearance via low-density lipoprotein receptor-related protein1 (LRP1) may follow [320]. Iron overload also impairs microglia motility and promotes senescence and a proinflammatory phenotype [321]. In addition, Fe accumulation in AD brains may be related to IL-6 mediated upregulation of hepcidin [322]. Accumulation of Fe in AD at an early stage led Bush and colleagues (2019) to suggest that the complex pathological process of AD and loss of neurons can be described as a ferroptosis [304,323]. Indeed, lipid peroxidation is increased and GPX4 and GSH are reduced in AD brains [324]. In mice models, deletion of GPX4 led to AD-like degeneration in the hippocampus and cognitive decline, whereas overexpression of GPX4 in 5xFAD mice AD model did ameliorate cognitive impairment and neurodegeneration [325,326]. In a small clinical study, N-acetylcysteine (NAC), a GSH precursor and inhibitor of ferroptosis [327], slowed cognitive decline in some AD patients [328]. A central mechanism of ferroptosis induction in AD underscores the role of Fe in addition to lipid dysmetabolism in the generation of the characteristic AD pathology [329]. Hence, to prevent ferroptosis, not only Fe reduction is necessary, but also a functional defense system with enough GSH and adequate supply of selenium for GPX4 synthesis.

Based on the hypothesis that Fe accumulation is an important contributing factor in the pathogenesis of AD, therapeutic approaches to decrease Fe overload in AD by Fe chelation have been tried [330]. In an early small study of AD patients, Fe chelation with deferoxamine reduced the rate of decline of daily living skills [331]; however, this agent poorly crosses the BBB. Deferiprone, which passes the BBB, has shown more promising results in a mouse tauopathy model [332], and reduced the burden of amyloid and p-tau in a rabbit model [333]. Several Fe chelators show effect in animal AD models, but only a few have been subjected to clinical trials [304]. Clioquinol and other 8-hydroxyquinoline derivatives have been shown to remove Fe from binding sites, i.e., amyloids plaques, in experimental studies in vitro and in vivo, and retard the amyloid plaque progression [304]. Although clinical studies on quinolines did not show clear clinical effect, post-hoc analyses have claimed proof of principle [334]. As cognitive decline in AD patients probably is driven by several pathophysiological mechanisms, metal chelation monotherapy is not expected to be a sufficient approach for treatment of AD.

**ALS** is a rare neurodegenerative disorder where progressive muscle wasting invariably leads to death in respiratory failure within on average three years from diagnosis. Diagnostic criteria include progressive motor impairment and presence of upper and lower motor dysfunction [335]. The cardinal neuropathological findings in ALS are nerve cell atrophy in the anterior spinal cord, reactive gliosis in the anterior horns, phosphorylated neurofilament aggregates in anterior horn axons and presence of protein aggregates with various types of cytoplasmatic inclusions [336]. Brain MRI shows bilateral hypointensities on SWI at the precentral gyrus [132] corresponding to the motor cortex, and in some cases frontotemporal atrophy. Other MRI and histopathology studies confirmed increased Fe in motor cortex but also in basal ganglia including caudate and subthalamic nuclei (Table 1) [133,134,280,337]; Fe deposits in the motor cortex were shown to be related to accumulation of Fe-rich microglia, at least in late disease stages [136].

The role of Fe in ALS pathogenesis is unclear, but possible dysregulation of Fe homeostasis precipitates as a possible mechanism explaining many features of the disease [338]. Presence of Fe in various ALS tissues has been studied since metal-induced oxidative damage was first suspected to contribute to nerve cell death in sporadic as well as in familial ALS. Kidney and liver tissues from ALS patients were then studied with neutron activation analysis and Fe concentrations were found significantly increased in ALS kidney compared to controls, and Fe and cobalt were also elevated in ALS liver tissue [129]. In a pioneering study on metals in ALS spinal cord, Kurlander et al. (1979) found significantly elevated concentrations of Fe together with lead and Cu [339]. A twofold increase in Fe concentrations was noted in ALS cervical spinal cord neuron nuclei [135] but not in capillaries and neuropil. These spinal cord studies were small, however elevated Fe levels in ALS spinal cords were later confirmed in a larger study from England where neutron activation analysis was used to evaluate Fe concentrations in lumbar spinal cord tissue from 38 cases and 22 controls [130]. Direct measurements of metals in ALS spinal cord using various methods have in different studies shown significantly increased concentrations of Fe, Cu, manganese, aluminum, selenium, Zn, and lead compared to controls (summarized in Table 2 in [131]).

Iron levels and Fe regulating proteins in body fluids from patients with ALS have been measured in several studies. Serum Fe concentrations seem to be increased in ALS [340] in parallel with increased ferritin and increased transferrin saturation where these changes correlate with disease duration [340]. Other studies have shown significantly higher plasma ferritin and lower transferrin concentrations in ALS patients than in healthy controls [341] and shown ferritin levels correlating to survival [342]. In a recent meta-analysis on Fe and Fe regulating proteins in ALS, however, no significant difference in Fe levels between ALS patients and healthy controls could be calculated, yet serum ferritin levels were found higher and transferrin levels lower in the overall assessment [128], indicating a possible use for ferritin as an ALS biomarker, and for transferrin as a potential treatment. CSF concentration of Fe may reflect Fe status in anterior horn cells of the spinal cord and in the CNS in general [343]. CSF iron was elevated in ALS, although at a non-significant level, in a Scandinavian study [344] which found significantly elevated CSF concentrations of other neurotoxic metals in ALS [131]. Patti et al. (2020) did not find significantly elevated Fe concentrations in CSF either in an Italian study, although differences in Fe ratios between bulbar and spinal ALS were noted [345].

Efforts to chelate Fe in ALS have so far not been successful [338]. Contemplations to correctly interpret these conflicting data concerning the role of Fe in ALS pathogenesis should also consider the fact that elevated Fe concentrations in neurons is a physiological phenomenon following nerve injury e.g., axonotomy [346].

## 6. Demyelinating Disorders and Neuroinflammation

MRI and histological studies have shown global alterations in Fe levels in the brains of MS patients in the white matter lesions and deep gray matter structures (reviewed in [347]). Oligodendrocytes are the most metabolically active cells in the brain probably owing to their role in myelination and contain an abundance of Fe-requiring enzymes that are important for oxidative metabolism. In addition, oligodendrocytes and myelin are rich in TfR, a potential source of the Fe accumulation in actively demyelinating MS lesions. Iron disturbances in lesions and deep gray matter seem not to be related, suggesting they have a different pathophysiological background [348]. From the deep gray matter structures, caudate nucleus, dentate nucleus, and globus pallidus exhibit the most striking changes in Fe content in MS [349,350]. Increased Fe concentration in these structures is associated with abnormal tissue integrity of adjacent white matter [351] as well as with disability and cognitive dysfunction [141,352,353,354]. More sophisticated MRI studies which assessed correlations of magnetic susceptibility, R2* transverse relaxivity, diffusion, and atrophy metrics suggested that individual deep gray matter regions may undergo differential alteration in Fe quantity and distribution such as its shift from oligodendrocyte to activated microglia, changes of chemical form or relative increase of concentration due to regional atrophy [349,355]. When taking tissue loss of the deep gray matter into account, several longitudinal studies suggested that while Fe concentration is increased due to overall tissue loss, total Fe content is unchanged or even decreasing in MS compared to controls [138,356,357]. MRI studies have consistently shown that Fe loss occurs in the thalamus, most strikingly in patients with secondary progression, and that this Fe loss is associated with clinical severity [138,139,141,356,358,359,360].

Increases in the Fe stored by macrophages and microglia are evident in a subset of early active lesions, which may indicate that Fe modulates the pathogenic process [361]. Iron content in active lesions may be the key marker distinguishing different MS immunopatterns. Iron deposits, when present in lesions, are in macrophages exhibiting signs of pro-inflammatory M1 activation state [362]. It was suggested that these immunopatterns may reflect disease heterogeneity and different demyelination mechanisms. Overall, Fe rims around demyelinating plaques composed of activated microglia and macrophages are specific for MS [363,364,365,366] and typical for chronic active lesions which increase in size for several years after their formation [143,367,368]. Iron rim lesions are associated with higher risk of conversion to MS in clinically isolated syndrome and with faster disease progression [142,369,370,371]. It was shown that in early lesions, most phagocytic cells are derived from CNS microglia while in later stages, these cells are replaced by macrophages recruited from peripheral monocytes [372]. Activation of microglia is apparent even in normal appearing white matter in MS patients compared to controls and it increases with disease duration [372].

A mass-spectroscopy study in post-mortem brain samples suggested that Fe deposition in MS is associated with increased levels of nuclear prelamin A recognition factor [373]. In pediatric MS with profound Fe deficit, patients stabilized after Fe supplementation. Authors speculated that oligodendroglial Fe deficiency leading to impaired ATP synthesis might have resulted in inefficient myelination and fragility of oligodendrocytes [374]. In line with this finding, mild association between decreasing transferrin saturation and increasing disability was shown in another study [375]. At the group level, serum Fe and ferritin are not significantly altered in most studies, but MS patients have lower total serum antioxidant capacity, higher hydroxyperoxides concentration, higher ceruloplasmin ferroxidase, and NOX activities suggesting ongoing oxidative stress [376,377,378]. Additionally, MS patients have decreased transferrin and lipocalin-2 in serum and CSF while higher lipocalin-2 levels are associated with increased deep grey matter Fe concentration [379,380]. In the CSF, several molecules involved in Fe metabolism, free hemoglobin, haptoglobin, and sCD163, were associated with higher cortical lesion load in newly diagnosed MS patients [381]. To this end, higher Fe concentration in the cortex was observed and found to be correlated to disease severity in MS [348]. Iron chelation therapy with deferoxamine or deferiprone has been tried with a limited success in MS [151].

A demyelinating disease akin to MS named Skogholt’s disease endemic in Odal County, Norway, is characterized by increased CSF Fe and Cu concentrations [382].

## 7. Iron Chelation and Other Means to Decrease Cerebral Iron

Experiences with chelation therapy in neurodegenerative diseases due to Fe deposition has here been discussed under each disease entity. This therapeutic approach has also been reviewed in a previous paper [151]. Some general principles are outlined here.

Iron chelation therapy has played a vital role in the management of patients with transfusional siderosis since the introduction of the parenterally administered chelator deferoxamine (Desferal) more than 50 years ago. The classical deferoxamine protocol involves the parenteral use of 25 mg/kg/day for 5–7-day-periods interrupted by therapy-free intervals. Obviously, such treatment is inconvenient and very stressful to patients. Therefore, an urgent need for efficient long-term oral Fe chelation therapy in outpatients has sparked the development of orally efficient Fe chelators suited for in-home treatment, such as deferiprone and deferasirox.

Deferiprone (Ferriprox) administered orally has now been used clinically in over 25,000 patients worldwide [383]. Deferiprone is rapidly absorbed in the gastrointestinal tract and normally appears in serum few minutes after oral administration. The main excretion route of the Fe-chelate is into urine. Deferiprone is a bidentate Fe chelator forming a 3:1 complex with Fe [384]. This chelator is likely to act intracellularly [385]. Experimentally, deferiprone has been shown to mobilize Fe deposited in brain [386].

Combined chelation therapy with deferoxamine and deferiprone are now often used in patients with siderosis [387]. It has been demonstrated that combined treatment with deferoxamine and deferiprone at doses lower than normally used, efficiently removed Fe from thalassaemia patients, indicating a potentiation of the Fe chelation efficiency with a combined therapy [388]. The advantages of combined chelation have precipitated the hypothesis that the two agents address different Fe-pools for chelation: deferiprone may chelate intracellularly whereas deferoxamine operates in the extracellular space. Thus, deferiprone is presumed to act as a “shuttling” agent [384].

A relatively new alternative in Fe chelation therapy is deferasirox (Exjade) [389]. This is an orally active tridentate chelator forming a 2:1 complex with Fe, the chelate being excreted via bile. Several clinical trials with this agent have reported its convincing efficacy in mobilizing Fe deposits in siderosis [390]. A daily oral dose of 20 mg/kg was considered as efficacious as the traditional parenteral deferoxamine treatment in controlling Fe stores. Maintenance therapy can be achieved by one daily dose.

It is of particular interest that deferasirox can be combined with deferiprone in the treatment of Fe overload [391]. This combination is preferable compared to the deferoxamine-deferiprone combination, due to the high costs associated with deferoxamine infusion. Monotherapy with either deferiprone or deferasirox may not always be optimally effective, especially not in patients with established cerebral Fe deposits. The oral deferoxamine-deferiprone combination has been reported to be safe and efficacious in thalassemic patients with suboptimal response to monotherapy [392].

Another principle of Fe mobilization is represented by the epigenetic modifier (histone demethylase inhibitor) drug GSK-J4, which lowered intracellular labile Fe levels in cultured dopaminergic neurons. GSK-J4 gave significant protection in SN of 6-hydroxydopamine rat PD model, presumably by inducing the Fe exporter ferroportin [393]. Indeed, ferroportin upregulation may reduce intracellular iron accumulation as has been shown in human erythrocytes [394]. Targeting ferroportin may be an interesting strategy to reduce brain iron accumulation.

## 8. Summary and Future Perspectives

The number of publications related to brain Fe metabolism has increased tremendously in recent years. It was stimulated by the development of novel MRI techniques, improvement of tissue metal mapping methods, and also the understanding of the ferroptosis mechanisms. Most neurodegenerative disorders are associated with deranged Fe metabolism in the brain or even systematically. It is, however, likely that mechanisms and consequences of brain Fe accumulation differ between individual diseases. In this work, we have summarized macroscopic and microscopic patterns of Fe accumulation in various neurological disorders. There are conditions with a relatively focal siderosis that typically takes place in a region with the most pronounced neuropathological changes, e.g., PD (SN pars compacta), PKAN, BPAN (globus pallidus, SN), and FRDA (dentate and red nuclei). In other disorders, including MPAN, MS, AD, and ALS, Fe accumulation is observed in the globus pallidus, caudate, and putamen, and in specific cortical regions in the latter two disease. Yet other disorders such as aceruloplasminemia, neuroferritinopathy, or WD manifest with diffuse Fe accumulation in the deep gray matter in a pattern comparable to or even more extensive than that observed during normal aging. The Fe deposition patterns described above are mostly derived from MRI studies. It was, however, suggested that the prevailing methodology for MRI assessment of Fe content, i.e., bulk estimate of mean Fe concentration (measured as R2* relaxation rate or magnetic susceptibility) in a manually or automatically segmented volume, is frequently biased by the atrophy of measured structure. Therefore, future MRI studies should optimally involve longitudinal measurements of Fe concentration in relation to tissue volume in regions of interest to ascertain whether the rise in Fe content exceeds its increased concentration due to volume loss. It was shown that MRI parameters reflect predominantly ferritin-iron; it is however not clear whether cellular ferritin-Fe content is linearly related to redox-active labile Fe pool and more research is needed to find out whether more Fe on MRI always implies harmful effects such as oxidative stress, neuroinflammation or ferroptosis.

Heterogeneity among disorders is apparent also in microscopic Fe distribution. In aging and in most neurodegenerative disorders, brain Fe deposits are present mostly in dystrophic microglia that may be accompanied by iron-laden, frequently perivascular macrophages, in disorders such as MS, WD, or NBIA group suggesting a role of immune cells in Fe accumulation. In other disorders, such as aceruloplasminemia and PKAN, most Fe deposits were observed in astrocytes implicating the role of BBB disturbance. Yet, in most disorders, Fe deposits are observed in various cell types including neurons and extracellular inclusions and it is not possible to classify disease types based on microscopic patters of iron distribution. To that point, it is interesting that defective TfR recycling was found to be a common abnormality in cellular models of genetic NBIAs, suggesting a joint abnormality in the regulation of Fe uptake in these diseases. Other works stressed the role of DMT1 upregulation and impaired recycling of cellular Fe by lysosomes. Interestingly, lipid metabolism disturbances were shown in several genetic NBIAs such as PLAN or MPAN. Speculatively, altered composition of phospholipid membranes in these disorders may render them more vulnerable to peroxidative membrane damage triggering ferroptosis cascade at a lower threshold. To date, it is not clear whether these mechanisms could also be relevant in sporadic neurodegenerations.

One of the major drawbacks in the research on brain Fe disturbances is that most studies target a single disorder and systematic studies comparing Fe metabolism in various conditions are lacking. Future research should seek biochemical markers associated with brain Fe deposits. It was shown that proteins related to Fe homeostasis, e.g., ceruloplasmin oxidase activity or transferrin isoforms, are altered in neurodegenerative disorders, but their relation to brain Fe levels is unknown. Lipocalin-2, a protein involved in immune reactions and Fe sequestering upon infection may be associated with deep gray matter Fe concentration in MS.

Iron disturbances appear to be associated with altered homeostasis of other metals. Brain Fe accumulation in aging, AD, and PD appears also to drive Cu dyshomeostasis, possibly causing Cu deficiency inside neurons that may contribute to neurodegeneration. It is becoming evident that research on Fe metabolism should involve examinations of its interaction with other metals. Furthermore, several recent independent studies showed that in NBIAs calcifications frequently accompany Fe deposits. In elderly persons, calcifications are commonly observed in small vessels of globus pallidus, a region with the highest Fe concentration.

Removal of Fe from cerebral deposits by chelation therapy is a possible, but difficult task. The conservative Fe chelation paradigm, which is based on shuttling Fe away from areas with local siderosis, is currently the prevailing approach. It may be beneficial in patients with aceruloplasminemia, late-onset PKAN, and possibly also PD; however, more experience is needed. Given the strong association between brain Fe levels and cognitive decline, it may be interesting to examine conservative Fe chelation in aging. An alternative to Fe chelation could be prevention of further brain Fe uptake, at least in those at risk due to genetic predispositions causing Fe accumulation. Blocking or downregulating TfR or upregulating hepcidin-ferroportin binding on brain barriers could be a successful strategy to limit further unnecessary Fe uptake into the brain. Targeted therapy toward other transporters or regulatory factors of ferroptosis such as NRF2 or GPX4 can also be an approach worth further studies. A current trend in pharmaceutical research is the development of multi-targeting drugs. Since inflammation, ROS, and ferroptosis are presumed to play a role in the case of Fe deposition, it is desirable that future drugs possess chelating, anti-ferroptotic, antioxidant, and anti-inflammatory properties.

## 9. Conclusions

Cerebral iron accumulation is common in several neurodegenerative disorders and in normal aging; it likely contributes to the pathophysiology of neurodegenerative changes in these conditions. Patterns of iron deposits in the central nervous system vary between different conditions, both at the macroscopic and microscopic level. It is likely that the role of iron in the pathogenesis and response to iron chelation therapy varies between individual disorders. Therapeutic strategies including prevention of brain iron accumulation and neutralization of excessive iron toxicity may prove beneficial in a subset of neurodegenerative disorders.

## Figures and Tables

**Figure 1 biomolecules-12-00714-f001:**
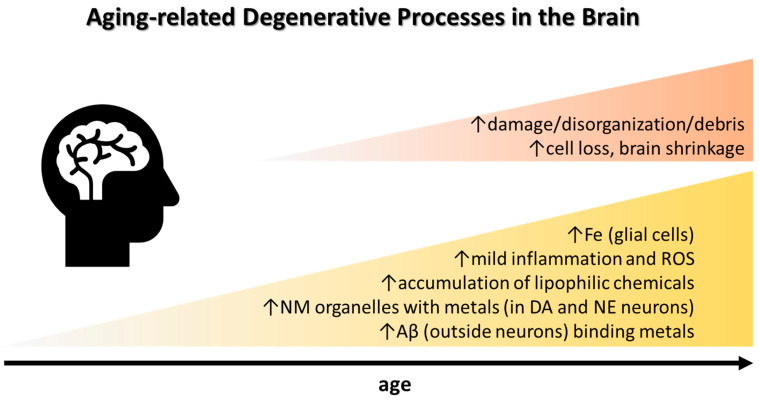
Common observations in healthy aging that may be accentuated in sporadic AD and PD. See text for details. Abbreviations: Aβ, amyloid-beta plaque; DA, dopamine; NE, norepinephrine; NM, neuromelanin; ROS, reactive oxygen species.

**Figure 2 biomolecules-12-00714-f002:**
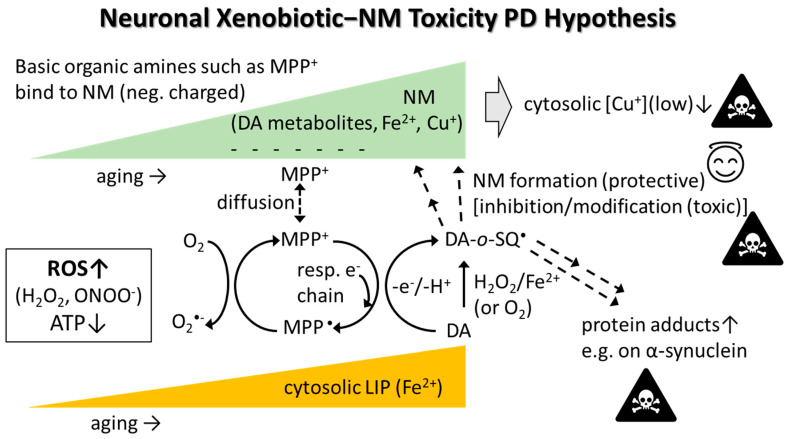
The neuronal xenobiotic-neuromelanin (NM) toxicity PD hypothesis. Negatively charged NM in SN attracts metal ions and positively charged xenobiotics such as basic organic amines (e.g., MPP^+^ and paraquat (PQ^++^)). Such organic xenobiotics diffuse into the cytosol and/or mitochondria where they redox-cycle, oxidize DA directly or through ROS production, and subsequent Fenton chemistry/autooxidation related to increased ferrous iron in the cytosolic labile iron pool (LIP). DA oxidizes into DA-*o*-SQ^•^ and further into various DA metabolites (e.g., DA-quinone, aminochrome, 5,6-dihydroxyindole) that oligomerize into NM. NM formation is generally seen as a protective process since reactive DA metabolites are removed. However, NM Cu sequestration lowers the already low cytosolic Cu levels which could be toxic (e.g., leading to less cytosolic Cu/Zn-SOD). Xenobiotics may also exert toxicity if they somehow modify the NM structure which could alter the redox activity of bound metal ions and facilitate ROS production. Some DA metabolites, e.g., DA-o-SQ^•^, DA-quinone and DOPAL (can be formed enzymatically), react with proteins including alpha-synuclein, forming toxic protein adducts (adducts may cause misfolding, and misfolded proteins tend to aggregate). If O_2_^•−^reacts with nitric oxide (NO^•^), peroxynitrite (ONOO^−^; strong oxidant) is formed. Abbreviations: DA, dopamine; DA-*o*-SQ^•^, DA-ortho-semiquinone radical.

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
