# Peer review of "Cerebral Iron Deposition in Neurodegeneration"

_biomolecules, 2022, doi:10.3390/biom12050714_

Round 1
Reviewer 1 Report
This review is well documented in detail on "cerebral iron deposition in neurodegeneration".
In page 10, I recommend to cite and discuss recent papers as belows about the connections between WDR45 deficiency and iron accumulation.
Aring, L., Choi, E. K., Kopera, H., Lanigan, T., Iwase, S., Klionsky, D. J., & Seo, Y. A. (2022). A neurodegeneration gene, WDR45, links impaired ferritinophagy to iron accumulation. Journal of neurochemistry, 160(3), 356–375. https://doi.org/10.1111/jnc.15548
Lee, H. E., Jung, M. K., Noh, S. G., Choi, H. B., Chae, S. H., Lee, J. H., & Mun, J. Y. (2021). Iron Accumulation and Changes in Cellular Organelles in WDR45 Mutant Fibroblasts. International journal of molecular sciences, 22(21), 11650. https://doi.org/10.3390/ijms222111650
Author Response
Please find the response attached.

Reviewer 2 Report
This is a very comprehensive review on the role of iron accumulation in NBIA diseases.The authors extensively describe the role of Fe accumulation in various neurodegenerative diseases and the mechanisms
that are altered by metal dehomeostasis. In my opinion, a more detailed explanation between iron accumulation and
neuroinflammation markers would be desirable.
Author Response
Please find the response attached.

Reviewer 3 Report
The work by Dusek and collaborators reviews available data on the levels of iron in different regions of the brain in various neurodegenerative conditions. Many reviews on this topic appeared recently in different journals, and this one differs from the other ones in the effort to cover all the neurodegenerative disorders in which iron is involved, including the various and complex NBIA forms. Its bibliography is extensive. The work is mainly clinically oriented, well written and well balanced.
- The summary and future perspectives give a summary of the locations of iron accumulation in the various disorders and the tools to study the accumulations. Surprisingly there is no effort for biological/biochemical interpretation of the data, ferroptosis is not even mentioned and so are the abnormalities of lipid metabolism found in forms of NBIA.
- in the paragraph on Hereditary Ferritinopathy it is stated: “Overall, results of this study suggest that abnormal FTL protein overexpression, aggregation and consequent proteostasis may be the primary cause of neurodegeneration while derangement of Fe metabolism may occur as a secondary event and that decreased T2/T2* signal in the grey matter could theoretically be caused by alteration of magnetostatic properties of aberrant Fe-FTL aggregates rather than by Fe accumulation per se.” This should be corrected, since there are no studies on the magnetostatic properties of the aggregates, while there are studies showing that ferritins containing the abnormal FTL are less efficient in forming an iron core.
- The two figures may be improved by explaining the acronyms used (NM, DA, NE, etc..) to facilitate the readers. Fig 1 may be made more graphical and appealing.
- Also in table 1 open PKAN, MPAN etc. should be open for easier understanding
Author Response
Please find response attached.
